# Effects of Nozzle Details on Print Quality and Hardened Properties of Underwater 3D Printed Concrete

**DOI:** 10.3390/ma16010034

**Published:** 2022-12-21

**Authors:** Jun-Mo Yang, In-Beom Park, Hojae Lee, Hong-Kyu Kwon

**Affiliations:** 1Department of Civil Engineering, Keimyung University, 1095 Dalgubeol-daero, Dalseo-gu, Daegu 42601, Republic of Korea; 2Korea Institute of Civil Engineering and Building Technology, Daehwa-dong, Goyang-si 10223, Gyeonggi, Republic of Korea; 3Department of Industrial and Management Engineering, Namseoul University, 91 DaeHakro, Seonghwan-eup, Cheonan-si 31020, Chungnam, Republic of Korea

**Keywords:** 3D concrete printing, underwater 3D concrete printing, print quality, density, compressive strength

## Abstract

This study developed a 3D concrete printing (3DCP) system that can print not only in air but also underwater. This underwater 3DCP system is equipped with many distinct technologies, such as a technology to supply the printing material to the nozzle tip at a constant rate by detecting its amount in the printer hopper. Using the developed 3DCP system, the effect of nozzle details on underwater print quality and hardened properties was investigated. The straight-line printing performance underwater was evaluated using five nozzles: a nozzle without a trowel (Nozzel#1), a nozzle with fixed trowels attached to both sides (Nozzle#2), a nozzle with trowels attached to the back and both sides to constrain five sides (Nozzle#3), a nozzle with a three-sided trowel inclined by 30° (Nozzle#4), and a nozzle with a roof added to Nozzle#4 opening (Nozzle#5). Nozzle#4 yielded the best print quality and hardened properties. In addition, an underwater curved shape printing test was performed using Nozzle#4, the problems that occurred in this test were analyzed and solutions were suggested.

## 1. Introduction

Currently, with the digitization of the global industry, additive manufacturing (AM) is being adopted by many industries such as the medical, aerospace, and automobile industries. Its use is also spreading to the construction industry, where the adoption of digital technology is very slow. Three-dimensional concrete printing (3DCP) is one of the fastest emerging technologies in construction AM [1,2].

The extrusion method is the most studied and applied method among 3DCP technologies [3,4]. In this method, a rigid and viscous cement-based material is extruded through a nozzle by applying pressure. The nozzle is digitally and automatically controlled to move it continuously, and the material is printed out along the nozzle and deposited layer-by-layer.

The main advantage of an extrusion-based 3DCP technology is that it can cast concrete and construct structures without a formwork. This increases the number of degrees of freedom in design and enables complex shapes to be produced within a short construction period while minimizing labor and construction costs. In addition, it can potentially minimize the waste of construction materials and replace many dangerous physical works in the field, such as tasks at a height, with 3DCP equipment. The latter has the advantage of realizing a safe construction site [5].

Despite the many advantages of an extrusion-based 3DCP technology, contradictory limitations and obstacles have not yet been completely resolved [6]: (1) controlling the rheological properties of fresh-state concrete and ensuring the deposition stability of the printed product [7,8,9]; (2) the standardization of the mechanical property evaluation and modeling methods for additive parts in the hardened state [10,11]; (3) ensuring a durability that meets the specifications [11]; and (4) effective integration of reinforcement [12,13]. However, because many studies are being conducted to solve these difficulties worldwide, an automated construction technology that fully utilizes the advantages of 3DCP will be practically applied soon.

Another challenge in an extrusion-based 3DCP technology is to extend its application to oceans. Automated construction technologies, such as 3DCP, will play a very important role in the development of marine resources and the construction of a marine city to be developed in the not-too-distant future. AM underwater is considered to have more freedom of shape than in air because the effect of gravity is reduced using extrusion 3DCP [14]. Underwater 3DCP can be used in the maintenance of damaged underwater structures. In addition, although the production of formworks for underwater structures with complex shapes is limited depending on their use, such as breakwater and reef blocks, manufacturing optimal underwater structures using underwater 3DCP technology is possible [15].

This paper presents one step in the technology development process that can use an extrusion-based 3DCP technology underwater. A 3DCP system that can be used underwater as well as in air was developed, and its details are explained herein. Using the 3DCP system, straight-line printing tests were performed in air and underwater, the details of the nozzle were changed to improve the underwater printing (WP) quality and hardened properties, and the optimal printing conditions were derived. In addition, using the derived optimal printing conditions, a curve-shaped printing test was performed, and the quality was evaluated.

## 2. Development Methodology of 3DCP System

A 3DCP equipment should be designed to meet the essential criteria directly related to the final objective of printing concrete structures using a concrete mixture. Therefore, determining the complementary relationship between the mix design of the materials and the printing equipment is necessary. As depicted in Figure 1, the basic approach to 3DCP technology development is to design/manufacture printing equipment and design 3DCP mixtures that satisfy the main factors of pumping, printing, and building performance required for printing structures.

### 2.1. Pumpability

Pumpability is defined as the property of conveying mix materials using a pumping device during the entire process while maintaining their fresh properties, allowing the printing device to appropriately operate. However, the requirements for pumpability are contradictory. Generally, a 3DCP mixture should be flowable to ensure easy transport to the printer extrusion device. However, the printed material must be relatively nonfluid to ensure that the deposited material retains its shape [16]. Therefore, pumpability depends on several pump parameters such as the pumping distance, pumping device performance, and type and diameter of the conveying pipe. Pumpable mixtures should have low plastic viscosities and moderate yield stresses.

As concrete flows in the conveying pipe, a lubrication layer forms on the pipe wall, and coarse aggregates move to the center of the pipe. In general, higher pumping pressures result in better pumpability; however, material segregation of the mix materials becomes probable. Material separation, in turn, can lead to material clogging inside the pipe owing to the loss of the flow layer [17]. In this study, pumpability was considered as the ability to continuously transport a 3DCP mixture to the printer extrusion device while maintaining its fresh-state properties without causing material segregation under high-pressure pumping.

### 2.2. Printability

Printability is related to the behavior of a mixture transported from the pumping device and the ability to combine by interacting with the printer unit nozzle [3]. Printability can be divided into the extrusion and deposition abilities of a 3DCP equipment, and it is also an essential index to evaluate the initial behavior of a 3DCP mixture. The ability to combine is based on the interaction between the behavior of the 3DCP mixture transported from the pumping device and the nozzle of the printer device [18].

A study on another property called buildability showed that it can be evaluated by the degree of deformation in the underlying layer caused by a newly printed layer on a particular layer [19]. In this study, printability was defined as the ability of a fresh mixture to be continuously extruded from the printer nozzle while printing a subsequent layer immediately on the lower layer with an acceptable deformation range.

### 2.3. Buildability

Buildability is used to evaluate the ability of a 3DCP mixture to withstand the load of its upper layer without collapsing as well as bearing the pressure generated during the printing process. A 3DCP mixture should maintain the shape deformation within an acceptable range after printing. The acceptable range is known to have a significant relationship with the specific performance (nozzle shape) of the 3DCP equipment [20].

Generally, in architectural 3DCP, the applied layer thickness should be set as small (<10 cm) to control the shape deformation and limit the initial gravitational stress. Thus, the limit of the layer thickness has a significant relationship with the nozzle size and shape of the 3DCP equipment [21]. The shear stress owing to gravity must be lower than the yield stress of the cementitious material to maintain its shape [22]. In this study, buildability was considered as the ability to continuously print five or more layers, measure the height of each layer, and print layers within an acceptable range of layer sagging.

## 3. Underwater 3DCP System

### 3.1. 3DCP System

A 3DCP mixture used for WP is significantly different from typical concrete materials. Research on WP is scarce owing to the printer equipment and mix material characteristics. Based on the investigations conducted thus far, Mazhoud et al. [14] produced WP specimens (with layer width < 40 mm and printed height < 76 mm). They used a six-axis robot-type printer equipment to examine the possibility of underwater output and the characteristics of the WP mixture.

This study conducted experiments on the characteristics of the developed underwater printer equipment capable of printing underwater structures (within 300 mm) using a four-axis gantry-type printer equipment. These structures have a layer width of 60 mm or more and a layer height of 30 mm or more. Experiments were also conducted on the performance of 3DCP mixtures for WP. Figure 2 shows the conceptual model of the WP system developed in this study. It consists of two primary devices.

(1) Pumping device and mixture transfer tube: the pumping device depends on the characteristics of the mixtures (slump: 4–8 mm and slump flow: 150–190 mm) used in a typical mortar-based construction 3DCP and prints a building structure with a certain layer width (<30 mm) and height (<15 mm) [23]. In this study, the mixture chosen for printing concrete structures in water using the 3DCP equipment presented high viscosity and low fluidity. For WP, a high-power motor and a high-performance progressive pump were needed to transfer the high-viscosity and low-fluidity mixture to the 3DCP equipment.

In a general mortar-based 3DCP for construction, a high-pressure rubber hose (diameter: <25 mm) is used as the material transfer pipe. In this study, the mixture used for WP exhibited high viscosity and low fluidity. Moreover, material separation should be avoided when transferring the mixture at a high pressure. Therefore, a high-pressure rubber hose (diameter: 50–60 mm) was also used as the material transport pipe connected to the printing device, and it was flexible.

(2) 3DCP equipment: it is a four-axis gantry robot that uses the Cartesian coordinate system to control the printing equipment and print a structure. The core technology of the printer equipment is printing a structure while extruding the transferred mixture at quantitative and constant speeds. In addition, the amount of mixture transferred to the printer hopper is automatically determined to control the progressive operation of the pumps. Considering the characteristics of the mixture and the pumping device, a hopper that could hold the transferred mixture was installed in the printer equipment. For WP, a waterproof rotation servomotor was installed on top of the nozzle. Moreover, a square nozzle and trowels were installed at the tip of the nozzle to minimize the external force acting in the water during printing and to ensure the dimensional accuracy of the printed product.

#### 3.1.1. Pumping Equipment of 3DCP System

As discussed in the Development Methodology section, pumpability refers to using a pumping device to transport a 3DCP mixture that meets the performance of the final objective and is printable to the printer hopper via the transfer pipe [24]. In addition, a prerequisite is that material separation should not occur during the material transfer process. High-viscosity, low-fluidity WP mixtures capable of printing underwater structures (with layer widths of over 60 mm and layer thicknesses of over 30 mm) require a slump flow (110–125 mm).

Progressive pump and low-power motors used in general industrial sites cannot transfer high-viscosity, low-fluidity WP mixtures to the printer hopper. A high-power pump and driving device are required to transfer WP mixtures required for underwater printing. As shown in Figure 3, a high-power progressive pump (2L6 rotor-and-stator screw pump) manufactured by the P company is used to transfer a mixture to the printer hopper. The transfer pressure of the progressive pump was 25–30 bar, which was driven by a high-performance AC motor (15 kW) and reducer (20:1).

#### 3.1.2. Printing Equipment of 3DCP System

The 3DCP equipment developed for this study was a four-axis gantry robot-type printer, which is shown in Figure 4. For operation control of the 3DCP equipment, the Xmotion series of LS Mecapion, Daegu, Republic of Korea, was used. Each axis was operated by a servomotor and its speed reducer, which controlled the movement speed, movement distance, and rotation angle of the axis. The operation of the servo motor of each axis was controlled automatically by interworking with a motor drive using a programmable logic controller (PLC) program, which is commonly used in automation processes.

Linear guide rails were used for the X- and Y-axes, and ball screws were used for the Z-axis. The nozzle rotation axis (axis A) was used to rotate a square nozzle using a spur gear. In addition, for WP, the rotating servomotor was equipped with particularly manufactured waterproof housing.

The mixture for WP had the characteristic that it did not flow under its own weight owing to its high viscosity and low fluidity. By installing a pressure plate inside the hopper, an appropriate pressure was transmitted to the mixture, which was subsequently extruded to the nozzle tip. An extrusion motor was mounted on the same concept as the spindle shaft used in machine centers for cutting [20]. Table 1 and Table 2 list the specifications of the linear/rotary guide and servomotor used for each axis.

When the printer equipment operated, the Z-axis received a large load, and two servo motors were arranged on it to minimize the vibrations occurring during acceleration/deceleration. These servomotors were controlled by synchronization. Their positions were controlled similar to that of a single servomotor.

A 3D printer for construction generally uses a circular nozzle to print a 3D shape; therefore, a rotation axis is not required. The developed 3DCP equipment uses a square nozzle and fixed side trowels; therefore, a rotating shaft was essential for printing 3D shapes. The use of square nozzles and side-trowels has various advantages over that of round nozzles, which have been introduced in academia and are widely applied in commercial printing equipment overseas [20,23,24,25,26,27]. As mentioned previously, the nozzle rotation shaft was driven by a spur gear. The rectangular nozzle installed on the rotating shaft printed a 3D shape in conjunction with the X and Y axes along the edge of the 3D shape. Figure 5 shows the extrusion device, which is the core module of the 3DCP equipment. It has many advantages in solving various problems that occur during printing of a structure:
(1)The transferred mixture has the characteristic that it does not flow well under its own weight. Actuation of the pressure plate by a spring transmits an appropriate pressure to the mixture. Thus, when the extrusion motor is operated, the printing material is extruded to the nozzle tip at a constant speed using a screw auger.(2)The sensors (photo sensors) connected to the pressure plate detect the amount of mixture transferred to the enclosed space of the printer hopper. The sensors consist of an upper limiter and a lower limiter. If the amount of mixture supplied is large, the pressure plate rises and touches the upper limiter, sending a stop signal to the pump. If no mixture is supplied, the plate moves down, touches the lower limiter, and sends an activation signal to the pump. Sending a signal in this way controls the operation of the progressive pump automatically.(3)It buffers the pulsation problem of the pumping device and the explosion problem of compressed air bubbles, which intermittently occur when transferring the mixture to the print hopper.

### 3.2. 3DCP Operation System

#### 3.2.1. Control Module of 3DCP Equipment

The developed control system used an MXP2.0 (CNC) motion controller supplied by Company LS. This module supports the EtherCAT communication and various topologies. In addition, it controls servo motors for up to 32 axes and 9 axes with simultaneous commands. It supports PLC and G-code programs and computer numerical control functions [28]. Therefore, a structure can be directly printed using a simple program modification process by uploading the G-code program generated from a typical printer slicing package to the control system. Figure 6 shows the system configuration and architecture of the motion controller.

In the structure of the control system, RMC Advanced Visualizer (RAV) is a function that interworks with the PC while editing and managing a human–machine interface (HMI). RMC Automation Studio (RAS) edits and manages PLC programs to control the servo motors and sensors of each axis. It also modifies and manages the parameters responsible for the position control of each axis.

The printer system control module shown in Figure 7 is divided into equipment operation and control units. The equipment operation part automatically controls the printer equipment by the HMI of the PC and manually controls the pumping equipment. The equipment control unit comprises a CPU and PLC module, motor drive, communication module, power module, and circuit breaker for controlling the printer and pumping equipment.

#### 3.2.2. HMI (Human Machine Interface) of 3DCP Operation

The HMI for the equipment operation in the control system was developed using RAS and RAV programs and consists of various windows. Figure 8 shows a virtual interface for printing structures by the operation of the 3DCP equipment. A brief description of each interface is as follows:

(1) AUTO window: this interface prints structures using a 3D slicing G-code. A camera function displays the movement status of the nozzle and the progress of the G-code when printing a structure and checks the input status of the transferred mixture in the printer hopper. It manages the basic information of the parameters (e.g., moving speed, extrusion speed, layer thickness, number of layers, and material slump) required for printing a structure and displays the machine and work coordinates of the printer equipment. When printing a structure, the printer equipment is temporarily stopped to install the support; after the support is installed, printing is continued.

(2) JOG window: the working coordinates of each axis are adjusted by individually operating each axis of the printer equipment. Two cameras are used to check the equipment status while running each axis individually.

In addition, after setting each axis origin individually, the equipment state is examined by a function to return to the machine origin. In addition, in the equipment operation, a crucial function is to operate each axis individually for equipment maintenance after printing the structure.

(3) EDIT window: this function uploads the G-code generated from the printer slicing package to the control system and subsequently modifies, verifies, or saves it. The simple G-code for equipment testing is written manually and saved as a file. G-code verification involves debugging all lines simultaneously or debugging one line sequentially. In the verification, the G-code is described as a 3D shape, and the problem line numbers are indicated if there are algorithm problems in the G-code.

## 4. Straight-Line Printing Test

### 4.1. Printing Materials

The mortar mix for the WP test of the 3DCP system was derived based on multiple mix tests performed several times in advance, and the mixture proportions are summarized in Table 3 [15]. The specified compressive strength was 50 MPa and the water-to-binder ratio was 38.4%. As a binder, 90% type 1 ordinary Portland cement was used, and 10% undensified silica fume was used to fill the voids between cement particles and ensure high strength and durability. The results of the ASTM C150 [29] test showed that the used cement had a specific gravity of 3.15 g/cm^3^, a specific surface area of 3770 cm^3^/g, an initial setting time of 210 min, and 290 min for final setting. The results of testing silica fume according to ASTM C1240 [30] showed the SiO_2_ content was 91.6%, loss on ignition was 2.3%, 45 μm residue was 3.9%, and specific surface area was 204,000 cm^2^/g. Silica sand with 95.5% SiO_2_ content was used as the fine aggregate. To maintain a constant particle size of the fine aggregate and shape stability of the printed product, Nos. 3, 6, and 7 silica sands with sizes of 1.2–2.4 mm, 0.25–0.70 mm, and 0.17 to 0.25 mm, respectively, were used. To improve the printability and buildability of the 3D printed concrete and provide an anti-washout performance underwater, a viscosity-modifying agent (VMA) produced by Dongnam was used with 2.0% water content [15]. The VMA dosage was determined through the anti-washout test and buildability test on the printing concrete with the VMA dosage variable. A powder-type methylcellulose VMA having a density of 0.75 ± 0.05 g/cm^3^ and a solid content of 97.0 ± 2.0% was used. To smoothly extrude and print the mix mortar using the 3DCP system, a light brown liquid polycarboxylic acid-based high-range water-reducing agent (HRWRA) with a solid content of 30.1 ± 2.0% and a density of 1.07 ± 0.1 g/cm^3^ was used. The HRWRA content for each printing test is listed in Table 3, and 0.5%–0.8% binder was used.

Mortar mixes based on the mixture proportions in Table 3 were used and printed using the 3DCP system similarly and in the same order in all six types of print tests. This reduced the change in the mechanical properties of 3DCP by the age of the mix material as much as possible [31]. After mixing each mortar, a slump-flow test according to ASTM C1437 was performed twice using the mortar immediately after the end of mixing and the first printed mortar to measure consistency [32]. When the hydration of cementitious particles is not considered, buildability and pumpability are closely related to the slump flow of a material [23]. The HRWRA content was adjusted to ensure the slump flow immediately after mixing was 110–120 mm. This was because if the slump flow is extremely small, clogging of the pump and poor printability may be caused, whereas if the slump flow is extremely large, the buildability may deteriorate. Based on Table 3, the slump flow at the start of printing increases by approximately 4–8 mm compared to that immediately after mixing. This is probably because high pressure and heat are generated as the mortar passes through the pump, hose, and hopper, causing the HRWRA to react more actively [15].

### 4.2. Nozzle

In this study, five types of nozzles, shown in Figure 9, were manufactured to evaluate the print quality of underwater additive concrete based on the difference in nozzle details. All nozzles were designed to receive a material with a 60 × 60 mm square cross-section and print a 60 mm wide and 30 mm high rectangular cross-section. Nozzle#1 was the simplest shaped nozzle with a 60 × 30 mm rectangular opening and a reduced cross-section in a ratio of 1:0.188. Nozzle#2 had the same opening as Nozzle#1, double the section reduction ratio (1:0.375) as Nozzle#1, and fixed trowels with a size of 30 × 30 mm installed on the left and right sides of the nozzle tip. For Nozzle#3, the section was reduced in a ratio of 1:0.167, and in addition to 40 × 30 mm trowels on both sides, a 60 × 30 mm rear trowel was installed. For Nozzles#1 and #2, the 60 × 30 mm cross-section of the printed part was determined by the nozzle opening of the lower surface. In contrast, for Nozzle#3, it was determined by the 60 × 30 mm front opening. Nozzle#4 had the same conditions as Nozzle#3 except that the three-sided trowel was inclined at 30° instead of being vertical. Nozzle#5 was manufactured by installing an upper roof on the front opening of Nozzle#4 and increasing the length of the trowel on both sides to 63 mm. Nozzles#1 and #2 were manufactured by cutting, bending, welding, and plating a steel plate, and Nozzle#3, #4, and #5 were manufactured by 3D printing with a polycarbonate material. 

### 4.3. Evaluation Method

Straight-line printing tests of the five types of nozzles were conducted in air and underwater. After completing the air printing (AP) on a plywood installed on a water tank, as shown in Figure 2, the plywood was removed, and WP was conducted in the water tank. A part was printed in a linear shape with a length of 1 m, and all layers were printed in the same direction to maintain the same time difference between the layers (Figure 10). The nozzle movement speed was fixed as 2000 mm/min. Therefore, the printing time gap between the layers was approximately 68 s. The printing height of each layer was set as 30 mm, and four layers were deposited. When each of the five straight-line printing tests were performed underwater, the measured water temperature in the water tank was in the range of 17.2 to 24.8 degrees. In this study, it was regarded the water temperature difference in that range has little effect on the deformation of the printed part. The effect of water temperature on the deformation of the printed part has not been studied yet, and the research will be conducted in the future.

In the printing test for each nozzle, three rows were printed by both AP and WP. In all rows, the hopper spindle-shaft rotation speed (HSRS) was set differently between 9 rpm and 16 rpm to produce differences in the print output amounts and evaluate the quality of each printed part. The correlation between the HSRS and the print output rate (printing mass per minute) was derived by measuring the weight of the printed mortar for a certain period. Figure 11, confirms a linear relationship between them. Accordingly, HSRSs of 10, 12, 14, and 16 rpm discharge print output at speeds of 13.2, 15.3, 17.5, and 19.6 kg/min, respectively.

After four layers and three rows were printed, the print quality was evaluated. Qualitative print quality was evaluated by examining the surface defects, deformation degree, and dimensional consistency of the top layer of the deposited part [33]. Quantitative printability and buildability, such as changes in the width and height of a deposited part compared to the designed values, were not evaluated. This is because the interlayer time interval and appropriate printing material properties, such as the initial yield stress and elastic modulus, were not determined considering the size of the target structure. Therefore, the deformation of the lower layer due to the deposition of the upper layer was inevitably large, which had a significant influence on the change in the width and height of the entire deposited part. Moreover, a standardized method for quantitatively evaluating the print quality is not available worldwide.

A specimen was manufactured to measure the density and compressive strength by coring the deposited parts prior to hardening. As shown in Figure 12, a cylindrical coated paper mold with a diameter of 50 mm and a height of 100 mm, with both open top and bottom surfaces was vertically inserted into the deposited parts at approximately 30 min. After the deposited parts were made, and the remaining samples outside the mold were removed to separate the six specimens. Specimens manufactured in air were cured in air for 2 days with plastic sheets covered, and after demolding, they were placed in curing water. In the case of specimens manufactured underwater, after separating the specimens from the deposited parts, they were transferred to a curing water tank, which was removed after two days, demolded, and cured in water again.

The densities of the six cylindrical specimens at 7 and 28 d of age for each deposited part were measured according to EN 12390-7 [34]. The mass used for the density calculation was measured with a scale in a dry surface condition in air, and the volume was obtained by actual measurements. The compressive strengths of the cylindrical specimens used for the density measurements at 7 and 28 d of age were measured according to ASTM C39 [35]. The compressive strength test was performed using a 5 MN compression tester, and a load was applied at a rate of 0.25 MPa/s.

### 4.4. Evaluation Results

#### 4.4.1. Print Quality

Figure 13 shows the results of the straight-line printing. In the deposited part, the first digit of the name is the nozzle number, the second digit is the HSRS, and the last digit denotes AP or WP. For example, 1–12 WP deposited part is a test specimen printed in water at an HSRS of 12 rpm using Nozzle#1. Table 4 qualitatively scores print quality. The scores for each surface defect, deformation degree, and dimensional consistency are subjectively marked from 1 (good quality) to 5 (bad quality), and the average of the three scores is calculated as the overall score.

Overall, dimensional consistency was found to be good. In the case of WP, surface cracking occurred from the side and the surface quality was worse than that of air printing in the same HSRS [15]. Presumably, the cohesion on the surface of the printout, which is the contact surface with water, was reduced because the mortar that passed through the nozzle came into contact with water. However, further research on the cause is needed.

For Nozzle#1, a large change in the cross-section of the printout was obtained. The cross-sections of all layers were trapezoidal. This was because the size of the deformation differed according to the height of the layer as the printout was printed. Moreover, the printing discharge pressure and its own weight increased as it went down to the bottom of the layer when the printout was printed and started to be deposited on the lower layer. This phenomenon became larger as the HSRS increased, causing the printing discharge pressure and height of the layer to increase. In the case of WP, a surface crack was observed in 1–12 WP, and the degree of surface defects was normal only when the HSRS was increased to 14 rpm or higher. However, a large deformation was caused by the high printing discharge pressure.

In the case of Nozzle#2, the trapezoidal cross-sectional shape was eliminated by installing fixed trowels on both sides. However, in WP, the surface cracks were larger than those in Nozzle#1. This may be due to the differences in the material rheology; however, presumably, the cracks increased as the fixed trowel scratched the cracked surface. This phenomenon could be reduced by increasing the printing discharge amount. However, at HSRS 14 rpm or higher, the discharge amount was excessive, and overflow in the direction of the nozzle movement occurred. If this overflow phenomenon continues to accumulate, the cross-sectional size of the printout may change significantly, and in curved printing, the curved shape may be deformed. 

In the case of Nozzle#3, the overflow phenomenon of Nozzle#2 was eliminated by adding a rear trowel to the trowel on both sides. In addition, as the opening was formed on the front side, the surface cracking phenomenon was reduced during WP while confining on five sides, except for the opening in contact with water. However, owing to the high confinement, the pressure on the lower layer increased, deformation of the lower layer occurred significantly, and a cross-section higher than the opening height was formed, showing a convex surface.

In the case of Nozzle#4, the three-sided trowel of Nozzle#3 was inclined by 30° to reduce the five-sided confinement pressure during printing, thereby reducing the deformation of the lower layer and the convex surface phenomenon. However, as the confinement was lowered, the surface cracking phenomenon during WP became larger than that of Nozzle#3. Regarding the overall print quality score, Nozzle#4 showed the best result, and among the WP specimens, 4–14 WP showed the best score with 2.0 points.

In the case of Nozzle#5, the surface convex phenomenon could be removed by installing an upper roof on the front opening of Nozzle#4. However, owing to the additional confinement of the upper roof, the deformation of the lower layer increased, and the surface cracking phenomenon was significant during WP. In addition, the length of the side trowels had to be increased from 40 to 63 mm for the installation of the upper roof, and the length of this extended trowel may have an adverse effect on curved-shape printing.

#### 4.4.2. Hardened Properties

Figure 14, Figure 15, Figure 16, Figure 17 and Figure 18 show the density and compressive strength results of all specimens measured at 7 and 28 d of age. Overall, the density and compressive strength trends were similar. In addition, a good print quality implies a high density and compressive strength. Therefore, if material and equipment system conditions that can satisfy good print quality are derived, the hardened properties will also be satisfactory.

In almost all specimens, the density increased with increasing age. The additional production of hydrates by the hydration reaction apparently contributed to the density increase. The density of the WP specimen was slightly lower than that of the AP specimen because the cohesion between the cement paste particles was reduced and loosened owing to contact with water [15]. Nozzles#3 and #4 showed different results because the trowel confines on five sides as soon as the printout contacts water, maintaining cohesion and preventing looseness between the cement paste particles.

In all specimens, the compressive strength tended to increase with increasing age, and most specimens showed a tendency of compressive strength increase as the HSRS increased. An increase in the printing discharge pressure increased the printout robustness. In Nozzles#1 and #2, under the same HSRS, the WP specimen showed a lower compressive strength than the AP specimen. This is also owing to the decrease in the cohesion between the cement paste particles in water. In the case of the 2–12 WP specimen, which showed severe surface cracking, the compressive strength at 28 days was significantly reduced by 13% compared to the 2–12 AP specimen. Nozzles#3, #4, and #5 did not show this trend because of the increase in the compressive strength during WP with the nozzle opening moving forward and being constrained from five sides. However, in the case of 3–16 WP and 4–16 WP specimens, the compressive strength was lower than those of the AP specimens and even lower than that of the low-HSRS specimen. This is because the deformation of the lower layer was extremely large owing to the high printing discharge pressure and confinement force.

## 5. Curved Shape Printing Test

Based on the results of the straight-line printing tests with different nozzle variables, the optimal conditions for WP were derived, and a curve-shaped printing test was performed underwater. As the printing material, a mortar mixed with the same materials and in the same mixing proportions as in the straight-line printing tests was used. In addition, 0.5% HPWRA was used. As shown in Table 3, the slump flows immediately after the end of mixing and at the time of initial printing were 110 and 114 mm, respectively. The reason for the reduction in the amount of the HPWRA is the minimization of the layer deformation and improvement in the buildability while ensuring rheology such that there are no problems in pumping and printing. To realize curves of various curvatures within a 1000 × 730 mm rectangle, the shape shown in Figure 19 was designed, a G-code was programmed, and a printing test was conducted. Nozzle#4, HSRS 14 rpm, and nozzle movement speed of 2000 mm/min, which is the combination with the best printing performance in the straight-line printing tests, were applied, and five layers were deposited with a one-layer height of 30 mm.

The results of the curved WP are shown in Figure 20. As summarized in Table 4, the print quality is evaluated as three points. The surface of the straight part was good, whereas surface cracking was observed in the curved part, particularly on the outside of the curved part with a large curvature. As expected, the 3DCP equipment in this study changed the direction of the nozzle movement while rotating it. Thus, when the nozzle was rotated, the distance inside the curve is short, and the distance outside the curve is long; thus, there was a difference in the amount of printing mortar discharged between the inside and outside of the nozzle. The deformation degree of the lower layer was good because the slump flow was minimized. The dimensional consistency of the straight-line part was good, whereas a width difference of the curved part with a small radius of curvature was observed. Moreover, the height of the outside of the curved part was lower than that of the inside. Therefore, the dimensional consistency was evaluated as 3.

After completing the curved WP, three φ50 × 100 mm cylindrical specimens were extracted from positions C1, C2, and S, which are shown in Figure 19. Similar to the straight-line printing tests, the specimens were manufactured by coring a coated paper mold vertically into the deposited part, and the hardened properties on 28 days of age after curing in water were measured. Figure 21 shows the results for density and compressive strength. Similar to the print quality results, the density and compressive strength of the curved parts (C1 and C2) were lower than those of the straight part (S). This also is owing to the non-uniform material discharge caused by the nozzle rotation in the curved part. The nozzle with fixed trowels used in this study has limitations in solving this problem. As suggested by Khoshnevis [25], a trowel can be deflected at different angles (by computer control). Using the printed material can improve the output quality of the curved parts and create various non-orthogonal structures with smooth and accurate surfaces. Furthermore, printing materials that can prevent the reduction in cohesion in the concrete matrix during WP should be developed.

## 6. Conclusions

The developed 3DCP system detects the amount of printing material in the printer hopper and automatically controls the operation of the progressive pump such that the printing material can be extruded to the nozzle tip at a fixed constant speed. In this study, the technical details of the developed 3DCP equipment, which have many advantages, were presented. The main advantages are buffering the pulsation problem of the pumping device, having precise control of the extrusion amount using a pressure plate installed in the printer hopper, and an excellent surface quality and dimensional accuracy of the printed layer using the square nozzles. Using this 3DCP system, printing tests in air and underwater were performed to improve the performance of WP by changing the nozzle details. Straight-line printing tests were performed using five nozzles: a nozzle without a trowel (Nozzel#1), a nozzle with fixed trowels attached to both sides of the nozzle (Nozzle#2), a nozzle with trowels attached to rear and both sides to confine five sides (Nozzle#3), a nozzle with a three-sided trowel angle inclined by 30° (Nozzle #4), and a nozzle with a roof added to the Nozzle#4 opening (Nozzle #5). By evaluating the linear printing performance, it was found that in the case of printing underwater, the surface quality was worse than that of air printing in the same HSRS. Nozzle#4 reduced cracking on the underwater-printed surface and had the best print performance. In addition, the hardened properties of the underwater-printed surface were good. Especially, 4–14WP showed the best print quality score with 2.0 points, which is the closest to 1 point indicating good printing quality, and the most high compressive strength of 56.0 MPa; therefore, Nozzle#4 was derived as the optimal nozzle for printing underwater A curved underwater printing test was additionally performed using Nozzle#4, and the printing performance was generally normal; however, problems of nonuniform material discharge and cracking of the outer surface in the curved part were observed. These problems could be solved by applying a trowel to the nozzle that moves automatically and can be applied from multiple angles. In addition, printed materials that can prevent the reduction in cohesion in the concrete matrix during WP should be developed.

## Figures and Tables

**Figure 1 materials-16-00034-f001:**
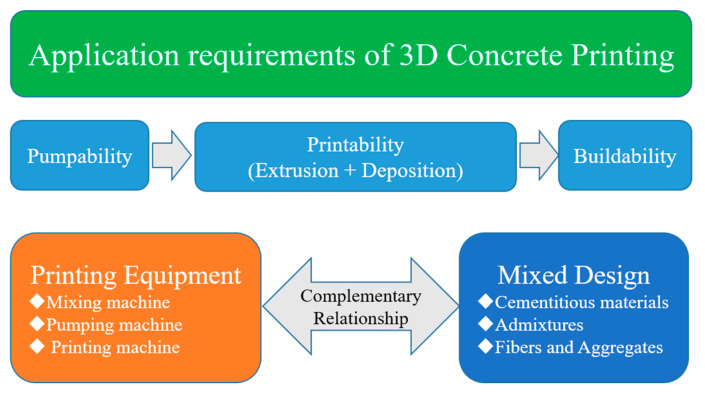
The performance of 3DCP during the printing process.

**Figure 2 materials-16-00034-f002:**
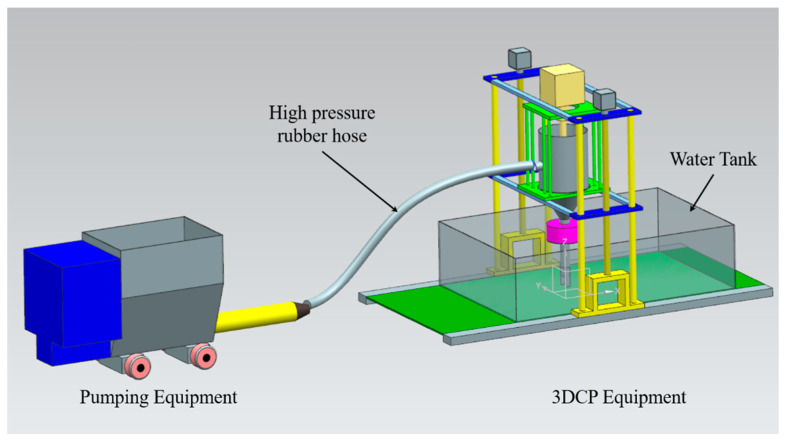
The concept model of 3DCP system for the underwater printing.

**Figure 3 materials-16-00034-f003:**
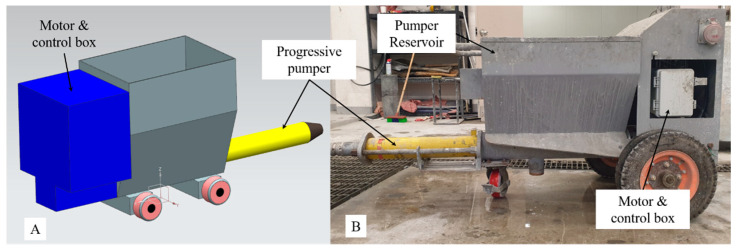
The pumping equipment of 3DCP system: (**A**) concept model; (**B**) developed machine.

**Figure 4 materials-16-00034-f004:**
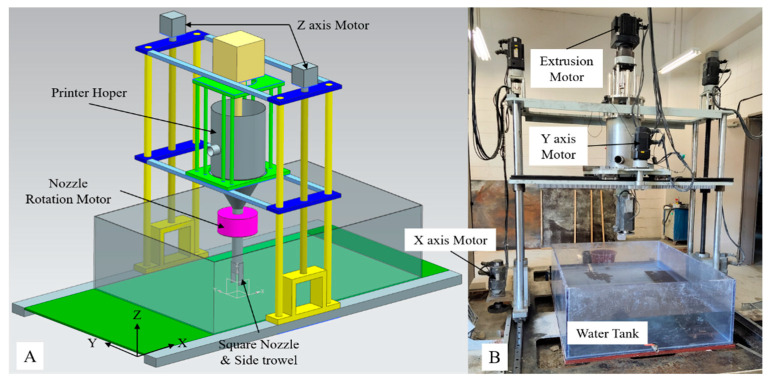
The printing equipment of 3DCP system: (**A**) concept model; (**B**) developed machine.

**Figure 5 materials-16-00034-f005:**
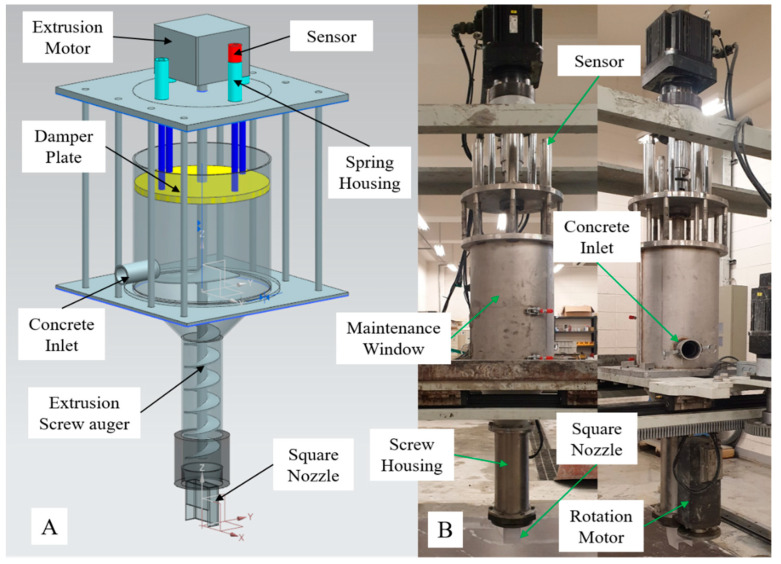
The extrusion module of 3DCP equipment: (**A**) concept model; (**B**) developed device.

**Figure 6 materials-16-00034-f006:**
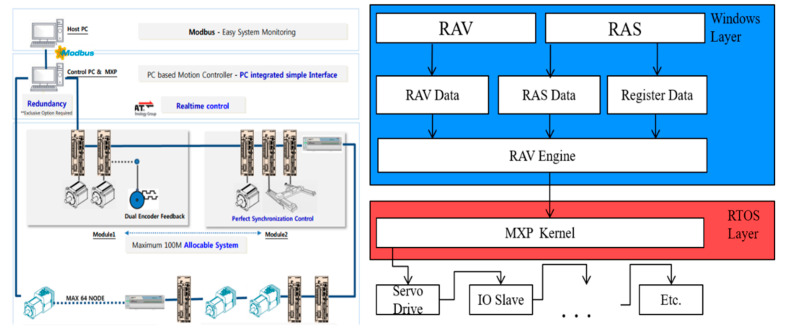
Configuration and structure of 3DCP control system [28].

**Figure 7 materials-16-00034-f007:**
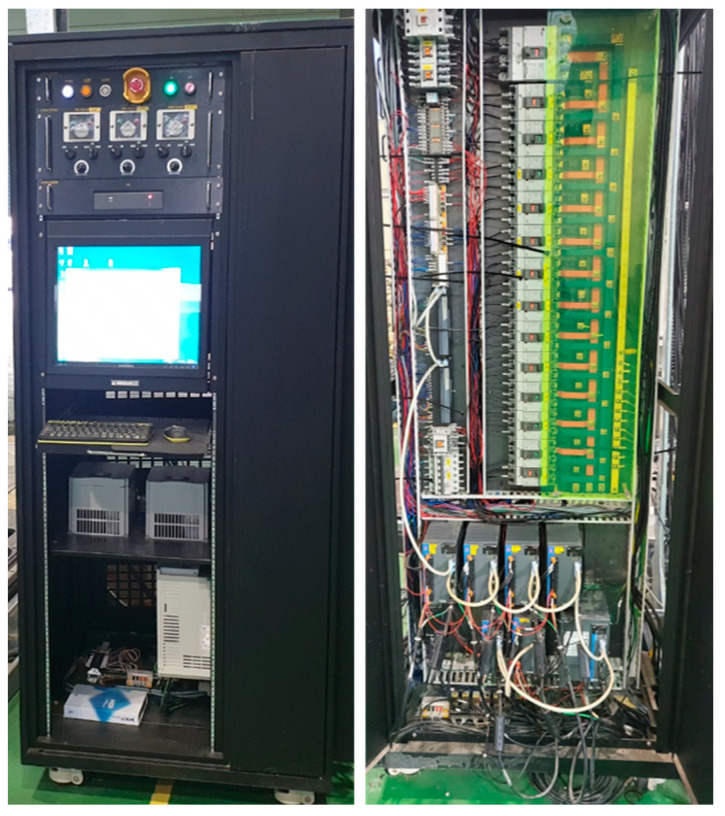
Control module of 3DCP system.

**Figure 8 materials-16-00034-f008:**
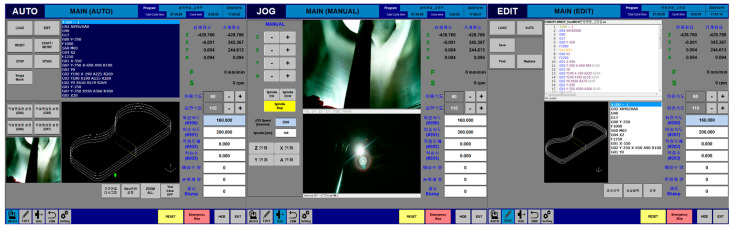
HMI of 3DCP equipment.

**Figure 9 materials-16-00034-f009:**
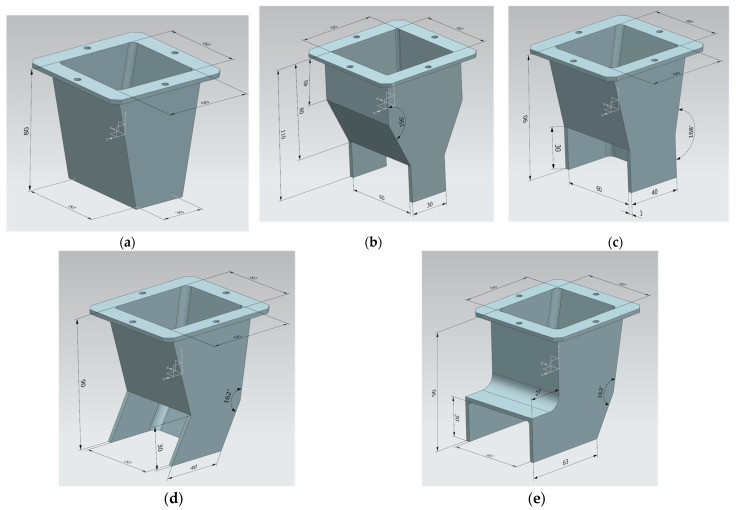
Types and details of nozzle (The unit of length is mm): (**a**) Nozzle#1; (**b**) Nozzle#2; (**c**) Nozzle#3; (**d**) Nozzle#4; (**e**) Nozzle#5.

**Figure 10 materials-16-00034-f010:**
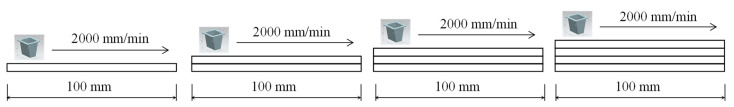
Designed printing test schedule for straight line.

**Figure 11 materials-16-00034-f011:**
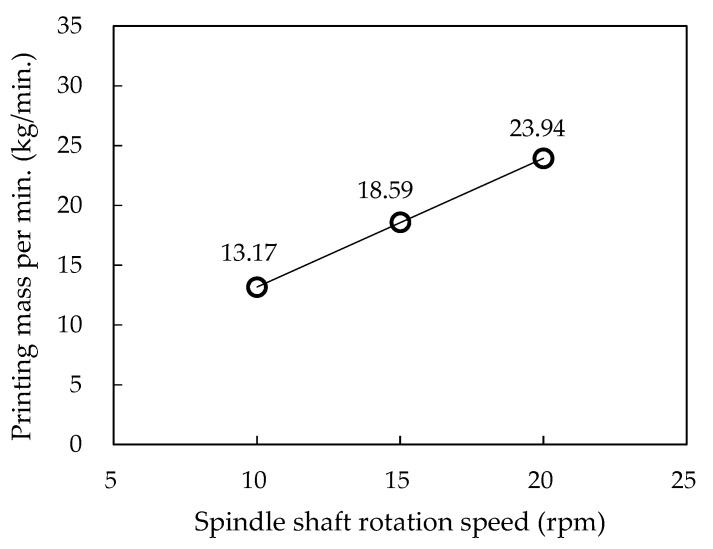
Relationship between spindle shaft rotation speed and printing mass per min.

**Figure 12 materials-16-00034-f012:**
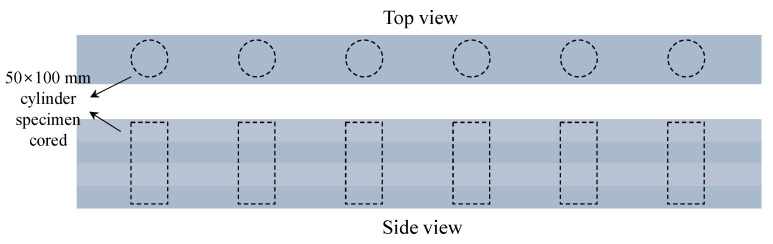
Coring six 50 × 100 mm cylinder specimens.

**Figure 13 materials-16-00034-f013:**
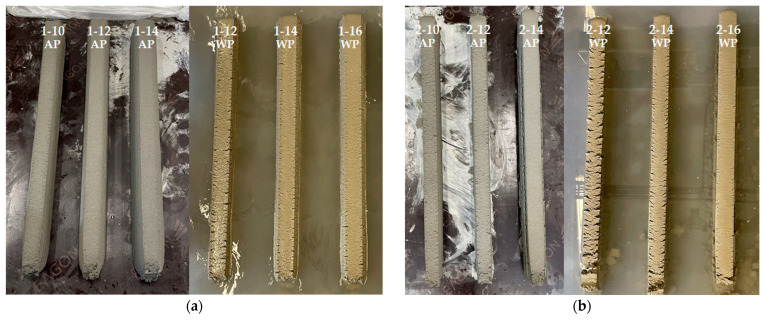
Results of straight-line printing: (**a**) Nozzle#1; (**b**) Nozzle#2; (**c**) Nozzle#3; (**d**) Nozzle#4; (**e**) Nozzle#5.

**Figure 14 materials-16-00034-f014:**
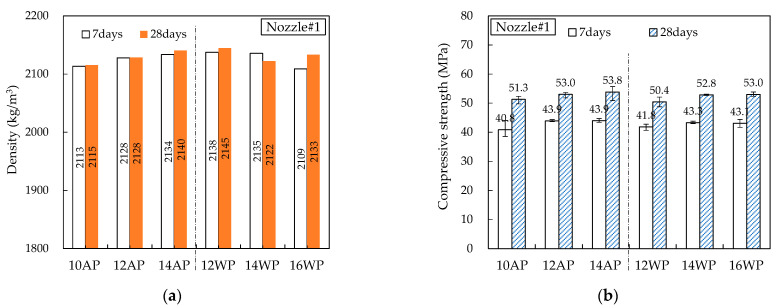
Hardened properties of Nozzle#1 specimens: (**a**) density; (**b**) compressive strength.

**Figure 15 materials-16-00034-f015:**
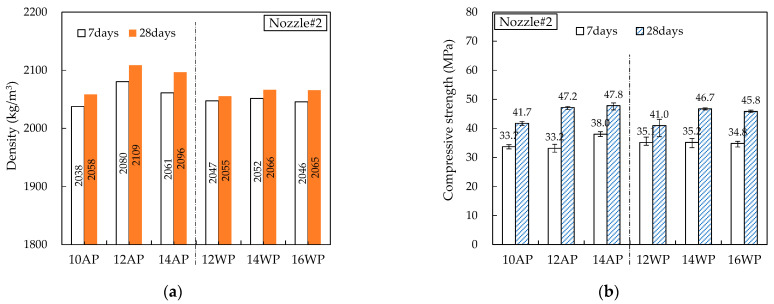
Hardened properties of Nozzle#2 specimens: (**a**) density; (**b**) compressive strength.

**Figure 16 materials-16-00034-f016:**
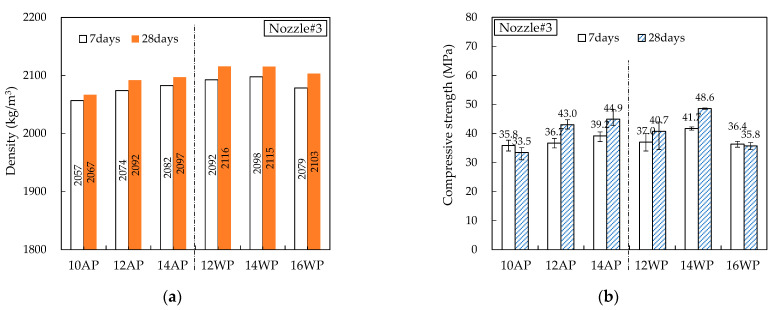
Hardened properties of Nozzle#3 specimens: (a) density; (b) compressive strength.

**Figure 17 materials-16-00034-f017:**
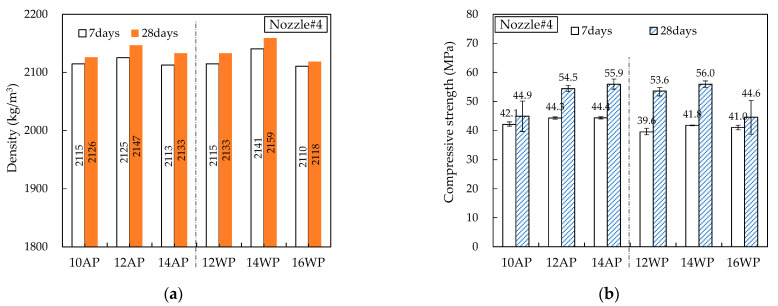
Hardened properties of Nozzle#4 specimens: (**a**) density; (**b**) compressive strength.

**Figure 18 materials-16-00034-f018:**
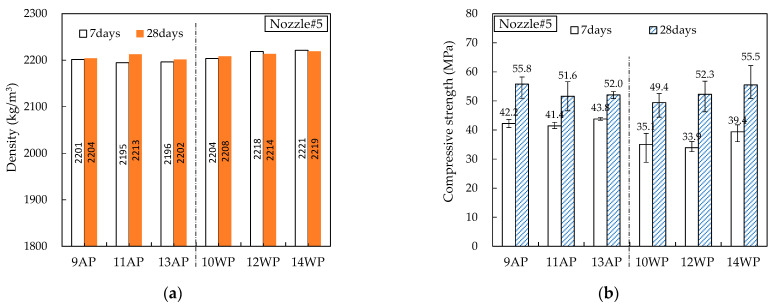
Hardened properties of Nozzle#5 specimens: (**a**) density; (**b**) compressive strength.

**Figure 19 materials-16-00034-f019:**
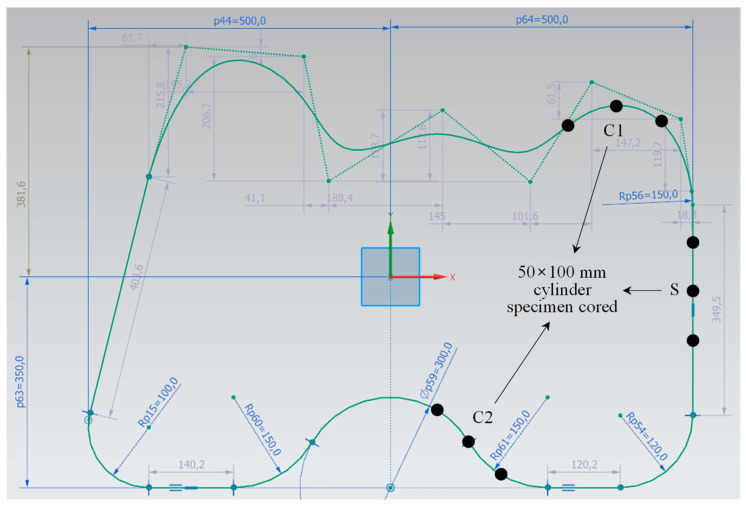
Designed curved shape for printing test.

**Figure 20 materials-16-00034-f020:**
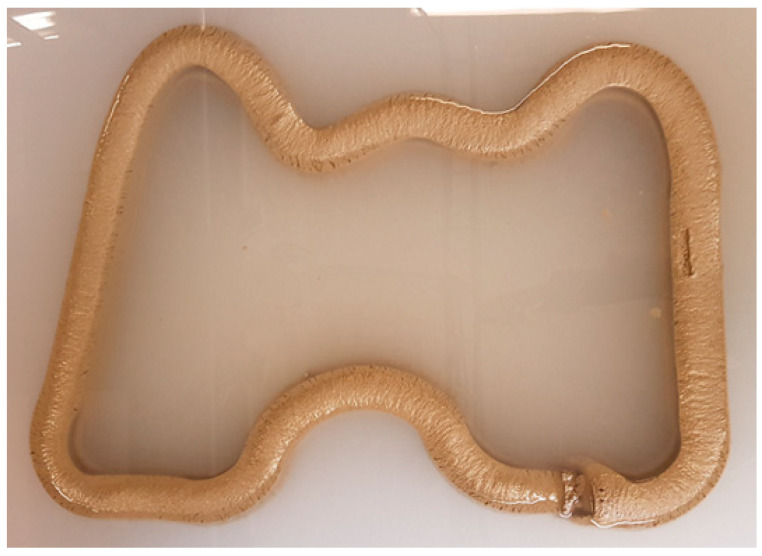
Result of curved shape underwater printing.

**Figure 21 materials-16-00034-f021:**
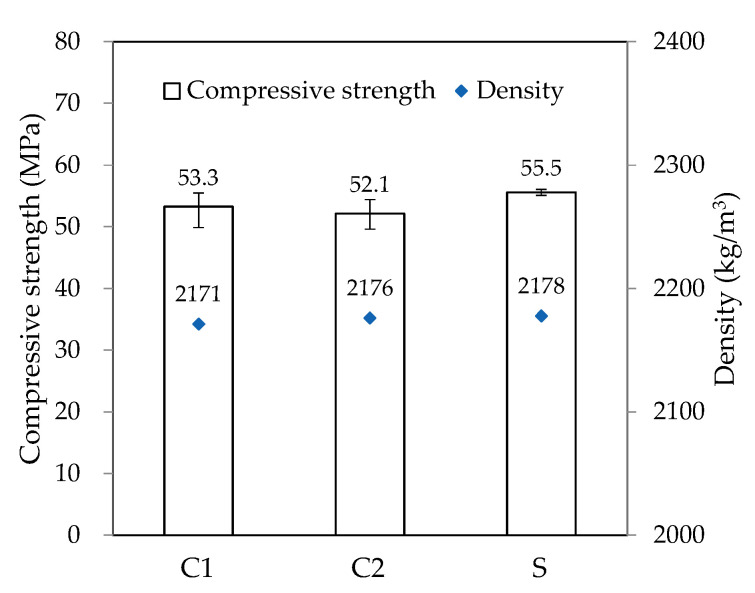
Hardened properties of curved shape underwater printing specimens.

**Table 1 materials-16-00034-t001:** Specification of linear and rotation guide.

	Moving Distance	Drive Guide Type	Repetition Accuracy
X1-axis	2500 mm	Guide rail & Rack gears	+0.05 mm
X2-axis	2500 mm	Guide rail & Rack gears	+0.05 mm
Y1-axis	1200 mm	Guide rail & Rack gears	+0.05 mm
Z1-axis	1500 mm	Ball screw	+0.02 mm
Z2-axis	1500 mm	Ball screw	+0.02 mm
A-axis	-	Spur gear	+0.02°
Spindle axis	-	Screw auger	+0.01°

**Table 2 materials-16-00034-t002:** Specification of servo motor.

	Rated Power (W)	Rated Revolution per Min (rpm)	Rated Torque (N.m)	Brake	Encoder	Motor Reducer
X1-axis	1.5 K	3000	4.77	o	Serial 19 bit	10:1
X2-axis	1.5 K	3000	4.77	o	Serial 19 bit	10:1
Y1-axis	1.5 K	3000	4.77	o	Serial 19 bit	10:1
Z1-axis	1.5 K	3000	4.77	o	Serial 19 bit	10:1
Z2-axis	1.5 K	3000	4.77	o	Serial 19 bit	10:1
A-axis	0.75 K	3000	2.39	o	Serial 19 bit	10:1
Spindle axis	5.5 K	2000	26.35	o	Serial 19 bit	20:1

**Table 3 materials-16-00034-t003:** Mixture proportioning and test results of slump flow.

Printing Test		W/B (%)	Unit Weight (kg/m^3^)				Admixture (%)		Slump Flow (mm)	
			W	C	SF	S	VMA	HRWRA	(1)	(2)
Straight line	Nozzle#1	38.4	250	586	66	1310	2.0	0.7	120	128
	Nozzle#2							0.7	113	119
	Nozzle#3							0.7	111	118
	Nozzle#4							0.7	117	123
	Nozzle#5							0.8	114	119
Curved shape								0.5	110	114

Note: W—water; C—cement; B—binder; SF—silica fume; S—sand; VMA—viscosity modifying agent; HRWRA—high range water reducing agent; slump flow (1): tested using mortar after the end of mixing; slump flow (2): tested using the first printed mortar.

**Table 4 materials-16-00034-t004:** Evaluation of printing quality.

Printing Test		Specimen	Free of Surface Defects	Deformation Degree	Dimensional Consistency	Overall Print Quality
Straight line	Nozzle#1	1–10 AP	2	4	2	2.7
		1–12 AP	1	5	2	2.7
		1–14 AP	1	5	2	2.7
		1–10 WP	4	4	2	3.3
		1–12 WP	3	5	2	3.3
		1–14 WP	3	5	2	3.3
	Nozzle#2	2–10 AP	2	2	1	1.7
		2–12 AP	2	2	1	1.7
		2–14 AP	1	3	1	1.7
		2–12 WP	5	2	1	2.7
		2–14 WP	4	3	1	2.7
		2–16 WP	3	4	1	2.7
	Nozzle#3	3–10 AP	2	2	1	1.7
		3–12 AP	1	3	1	1.7
		3–14 AP	1	4	1	2.0
		3–12 WP	3	3	1	2.3
		3–14 WP	2	4	1	2.3
		3–16 WP	1	5	1	2.3
	Nozzle#4	4–10 AP	1	2	1	1.3
		4–12 AP	1	2	1	1.3
		4–14 AP	1	3	1	1.7
		4–12 WP	4	2	1	2.3
		4–14 WP	2	3	1	2.0
		4–16 WP	1	5	1	2.3
	Nozzle#5	5–9 AP	3	2	1	2.0
		5–11 AP	2	3	1	2.0
		5–13 AP	1	4	1	2.0
		5–10 WP	4	3	1	2.7
		5–12 WP	2	4	1	2.3
		5–14 WP	1	5	1	2.3
Curved shape			3	3	3	3.0

Note: 1—good; 3—normal; 5—bad.

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
