# Peer review of "Effects of Nozzle Details on Print Quality and Hardened Properties of Underwater 3D Printed Concrete"

_materials, 2022, doi:10.3390/ma16010034_

Round 1

Reviewer 1 Report

Manuscript ID: materials-2088352

Title: Effects of nozzle details on print quality and hardened properties of underwater 3D printed concrete

Reviewer Comments: Authors must respond to these comments in order to improve the paper.

1.      In Table 1, repetition accuracy column, include the abbreviation for “o”?

2.      Introduce a new section, i.e., Section 6, and incorporate information about issues encountered and precautionary measures taken during pumping concrete.

3.      The author stated that they used an algorithm in section 3.2.2, lines 279-281; in the same section, they incorporate the algorithm they used in their research.

4.      Add a high-resolution image in figure 8.

5.      Why the author did not include basic tests carried out on concrete

6.      On what basis did the author choose the VMA dosage?

7.      What type of HRWRA admixture was used in 3d concrete?

8.      On what basis were the nozzle size reductions made?

9.      Authors have experimented with underwater printing concrete without carrying out an investigation of its water absorption property why?

10.  Conclusion is should be more precise rather than discussion.

11.  Authors can graphical abstract to attract the readers.

12.  The author can include a methodology section that details the various stages involved in the development of 3D concrete.

13.  The authors explain how they ensured that the materials used in production had high viscosity and low fluidity in section 3.1, line 145.

14.  What is the percentage of wastage of concrete during pumping of concrete, and what precautionary measures did the author take to reduce the wastage and blockage of concrete in pumping?

15.  Line 221: "What type of sensors were used and what were the steps involved in the processing of raw data from sensors?" needs to be written in the paragraph.

16.  A comparison study is required in this article. The author can prepare a comparison table of various test results from past literature and compare them with their results. The following are a few articles for references.

doi.org/10.1016/j.jmrt.2022.01.045

doi.org/10.3390/ma14143839

doi.org/10.1016/j.conbuildmat.2022.129535 

Author Response

Please find an attached file.

Reviewer 2 Report

The article is devoted to the study of the influence of the nozzle shape of a construction 3D printer on the deformation of a printed product printed underwater and in air. The article describes the methods and materials well, the research itself was carried out at a high level, but I have a few questions for the authors. Please find them below: 
1) In table 3, it is not clear what Nozzle# — 1Nozzle#5 means. Alternatively, table 3 can be moved a little lower in the text.
2) Which a plastic material is used for nozzles?
3) What was the temperature of the water, how does it affect the deformation of the printed part?
4) In conclusion, there are no numerical estimates of the results obtained. It should be added why nozzle 4 is the most optimal.
5) The article should clearly indicate the contribution to the study area. Line 558: "In this study, the technical details of the developed 3DCP equipment, which have many other advantages, were presented."  -"...many other advantages" is better to specify which advantages.
6) The introduction section lacks an overview of existing underwater 3D printing systems.
7) Author's paper, "Comparison of Properties of 3D-Printed Mortar in Air vs. Underwater", has a rather similar structure. Indicate the novelty of this work compared to the previous one.
The list of references can be improvement, for example add some papers:
printing strategies for a 3D concrete printing:
https://doi.org/10.1016/j.jobe.2021.103599
about mechanical design (interesting examples of mechanical design 3D printers in appendix):
https://doi.org/10.3390/app12094514
application of underwater 3D printing in bioprinting:
https://doi.org/10.1021/acsami.1c24332
I hope that you find these comments constructive for refinement of your work. With all mentioned, I believe the work may be published after the major revision.

Author Response

Please find an attached file.

Round 2

Reviewer 1 Report

The revised manuscript can be accepted 

Author Response

We would like to thank you for your excellent comments which significantly improved the quality of our paper.

Reviewer 2 Report

Thanks to authors for answers for comments, but there are some questions to new version of the paper:
 1) The plastic material is polycarbonate, but it is common name, please write more concretely (PLA, PETG)
 2) "Authors believe that the water temperature difference in that range has little effect on the deformation of the printed part. The effect of water temperature on the deformation of the printed part has not been studied yet, and we plan to conduct research on this in the future". It would be good, if this text will be added to conclusion.
 3) In conclusion, explain what means print quality score with 2.0 points. In table 4 included some results for air printing which can added to the conclusion. For example, which nozzle is better for air printing and why.
 4) "In this study, the technical details of the developed 3DCP equipment, which have many other advantages such as buffering the pulsation problem of the pumping device, applying a square nozzle using a rotating shaft, and so on, were presented."
"..and so on" What else? List all advantages.

 With all mentioned, I believe the work may be published after the minor revision.
